

# Intelligent toy tracking trajectory design based on mobile cloud terminal deployment and depth-first search algorithm

Yang Zhang and Hu Zhang

Beijing AIQI Technology Co., LTD., Beijing, China

## ABSTRACT

The popularization of intelligent toys enriches the lives of the general public. To provide the public with a better toy experience, we propose the intelligent toy tracking method by the mobile cloud terminal deployment and depth-first search algorithm. Firstly, we construct a toy detection model *via* Transformer, which realizes the positioning of toys in the image through the refined region adaptive boundary representation. Then, using these detected continuous frames, we improve the toy tracking based on a depth-first search. Long-short-term memory (LSTM) constructs the continuous frame tracking structure, and the depth-first search mechanism is embedded to realize the accurate tracking of multiple targets in continuous frames. Finally, to realize the terminal marginalization of the proposed method, this chapter proposes a lightweight model deployment method based on mobile cloud terminals to realize the maintenance of the optimal machine state of intelligent toys. The experiment proves that our proposed target method can reach the world-leading level and obtain the mAP value of 0.858. Our tracking method can also perform excellently with a MOTA value of 0.916.

## INTRODUCTION

With the booming development of the smart toy market, trajectory tracking technology has become a crucial innovation point that captures consumers' attention. This technology enables realtime tracking of the movement trajectory of smart toys, providing users with a more intelligent, personalized, and highly interactive gaming experience. Consequently, it significantly enhances user satisfaction and deepens their reliance and trust in smart toys. More importantly, leveraging trajectory tracking technology for smart toys, we can develop intelligent toys that greatly benefit children's learning and growth. These toys not only entertain but also possess educational value, thus exhibiting tremendous practical application significance. The trajectory tracking of intelligent toys has distinct technical characteristics (*Akdeniz & Ozdinc, 2021*; *Moradi, Amiri & Ghanavi, 2017*). First, the trajectory tracking method of intelligent toys requires realtime acquisition, processing, and display of the location information of toys in real time to ensure realtime monitoring and interactive gaming experience. Secondly, accurate positioning information is the basis of

Corresponding author
Yang Zhang, 13683602101@163.com

intelligent toy track tracking, and positioning technology is required to provide high-precision location data indoors and outdoors (*Druga, Williams & Park, 2018*). Secondly, due to the portable type of toys, trajectory tracking technology must be designed with lightweight, small, and simple sensors and algorithms that can be easily integrated into various toys (*Luo, 2023*). Finally, smart toys may be used in different environments, including indoor, outdoor, different terrain, *etc*. The trajectory-tracking technology should have adaptability and stability (*Wang, Yin & Zhang, 2021*; *McStay & Rosner, 2021*; *Chen, Wang & Huang, 2006*). Therefore, factors such as high-precision positioning, changing scenes and complex path movement are difficulties tracking intelligent toys (*Lemma, Celik & Katzenbeisser, 2008*).

To deal with the above difficulties, many researchers focus on single-object tracking technology to conduct tracking research on intelligent toys (*Qian, Li & Xue, 2023*; *Yang, Lu & Wu, 2018*; *Delprino, Piva & Tommasi, 2018*). *Frossard & Urtasun (2018)* designed an end-to-end tracking method, which includes different network structures for processing point cloud and image data, to realize realtime target detection, information matching and linear optimization because both the detection and matching modules adopt the DNN (*Zhang, Zhou & Sun, 2019*). *Luiten, Fischer & Leibe (2020)* employed 3D reconstruction for the occlusion and improved the tracking. The MOTSFusion framework proposed by LutsFusion consists of two stages. *Cao et al. (2023)* designed a tiny-object detection and tracking method DT by SiamFC tracking network structure by adding improved HOG and Harris algorithms. Experiments show that the model assesses faster tracking speed and higher accuracy. *Pang et al. (2023)* proposed a target-tracking model by image block matching to overcome the inaccurate tracking caused by blurred and noisy images of high-speed moving targets. The experiment can conclude that this method performs better in high-speed motion tracking. In addition, some scholars (*Zhang, Wang & Zhang, 2023*) improve the tracking effect by enhancing the object detection algorithm.

1) However, the current approaches are predominantly terminal-based model architectures, which struggle to address the realtime and lightweight requirements essential for smart toys. Furthermore, these existing methods fail to achieve multi-scene trajectory tracking and realtime positioning in unknown environments. To address these limitations, this article proposes a novel smart toy trajectory tracking method called TTNet based on a mobile cloud terminal and depth-first search algorithms. The TTNet method leverages mobile cloud terminals' computing power and storage capabilities to ensure realtime performance and lightweight implementation. TTNet can maintain rapid response times even for resource-intensive tracking algorithms by offloading complex computations to the cloud. This allows for seamless tracking of smart toy movements, ensuring a smooth and engaging user experience.

2) Moreover, TTNet incorporates depth-first search algorithms to enable multi-scene trajectory tracking. By utilizing the search paradigm of depth-first search, TTNet can effectively explore and map unknown environments, enabling accurate realtime positioning even in unexplored scenarios. This capability significantly enhances the adaptability and versatility of the trajectory tracking system, making it suitable for a

wide range of real-world applications. Main contributions are as follows: We propose an intelligent toy detection model based on Transformer, which realizes precise positioning of toys by designing an adaptive boundary regression model.

3) We propose an intelligent toy trajectory tracking method based on depth-first search, which constructs the inter-frame relationships to achieve realtime tracking.

4) Using our approach, we propose a lightweight model deployment method based on mobile cloud terminals to enable mobile applications.

## RELATED WORKS

### Object detection technology

Object detection refers to a class of algorithms that extract key targets from the target area of interest in images, videos, or point clouds. Obtaining reliable and accurate target detection results is of great significance to the subsequent target track tracking, so the first prerequisite for researching target track tracking methods is to propose an accurate target detection algorithm.

In 2012, *Krizhevsky, Sutskever & Hinton (2012)* proposed the AlexNet model. Since then, detection can be divided into two- and single-stage. The two stages mainly complete the image target detection by obtaining the detection candidate region and target classification detection. The region to be detected can be obtained through selective search or a regional suggestion network. Faster R-CNN (*Ren, He & Girshick, 2015*) is the most representative method. A single-stage algorithm means that only one target feature extraction is needed to achieve target detection, among which Yolo (*Lan, Dang & Wang, 2018*) represents single-stage detection. *Ma et al. (2018)* designed a target detection model by full anchor training for road target detection. The test results on the KITTI dataset demonstrate that the model can identify small and medium objects and blocked objects on the road well. *Meng, Rice & Wang (2018)* proposed the adaptive candidate region adjustment layer and the adaptive confidence threshold selection method, which reduced the calculation parameters of Faster R-CNN and improved the model detection accuracy. *Chen, Yang & Kong (2017)* proposed an obstacle-detection algorithm for road scenes. By replacing VGG16 of the original network with ResNet and improving the area suggestion network, the precision measurement accuracy of the original algorithm is effectively improved.

3D data is widely used in environment sensing tasks because it can provide more abundant scene information. *Yang et al. (2020)* use Distance FPS and Feature FPS. The features are inputted to the candidate generation layer and a 3D detection frame is generated through the anchor-less frame detection head. The graph-based method realizes the target detection of 3D point cloud data through the graph neural network. *Shi & Rajkumar (2020)* represented the point cloud with a graph, performed feature extraction through multi-layer perceptron, and predicted the target detection frame through the iterated point cloud features. Some scholars also proposed target detection with the original point cloud as input by focusing on cloud timing information and adding a multi-stage feature extraction network (*Miao et al., 2021*; *Noh, Lee & Ham, 2021*).

## Target trajectory tracking technology

Multi-target tracking is to explain how to conduct association for the same target in different frames. First, a target detector is used to detect each frame of the Video, and then the detection result is sent to the tracker; the tracker will allocate a unique ID to the detection target, and the targets with the same ID in different frames represent that they are the same target. *Bewley et al. (2016)* first employ the Kalman to regress the region of the object for the next step and then calculate the IoU for the predicted result and the detection results in the next frame. If there was no corresponding prediction result in the detection result, a new ID was assigned to the detection result. However, if an object is blocked, the algorithm will lose the object. *Wojke, Bewley & Paulus (2017)* introduced a feature extraction network to calculate the feature of the frame and then jointly used the feature and IoU to match the detection frame. Based on CenterNet, *Zhou, Koltun & Krahenbuhl (2020)* input the picture of the current time, the picture of the previous one and the hot spot map, perform downsampling and summing, respectively, and then obtain the heat map of the current frame through upsampling through convolution and batch normalization layer (BN): confidence, displacement and other information. Based on CenterTrack, *Zhang, Wang & Wang (2021)* added the Re-ID branch to make it capable of object detection and tracking. They would make predictions for each pixel, predicting whether it is the object's center and the Re-ID feature of the image area centered on it.

# INTELLIGENT TOY TRACK TRACKING ALGORITHM BASED ON MOBILE CLOUD DEPLOYMENT AND DEPTH-FIRST SEARCH

Aiming at the problems of background interference and difficult multi-target concurrency in intelligent toy tracking, we propose the intelligent toy tracking method based on a mobile cloud terminal and depth-first search (DFS). A transformer, long short-term memory network (LSTM), and deep search algorithm are adopted to develop an intelligent toy track-tracking method that integrates intelligent toys and multiple scenes.

## Intelligent toy detection model based on Transformer

First, we regard the intelligent toy detection model by Transformer as the toy target locator. Its codec structure has a strong receptive field, whose network is in Fig. 1. The Transformer model exhibits significant advantages in object detection tasks. With its unique self-attention mechanism, the Transformer can establish direct connections between elements at different positions in the input sequence, effectively capturing long-range dependencies in images. This capability enables the Transformer to more precisely identify object features in images, especially when it involves subtle differences between objects and their backgrounds and the interrelationships between multiple objects.

Furthermore, the Transformer's encoder-decoder architecture allows it to map image features directly to object-bounding boxes and categories, achieving end-to-end object detection. This concise and efficient approach simplifies the object detection, improving detection efficiency and accuracy. Notably, the Transformer can process larger images and
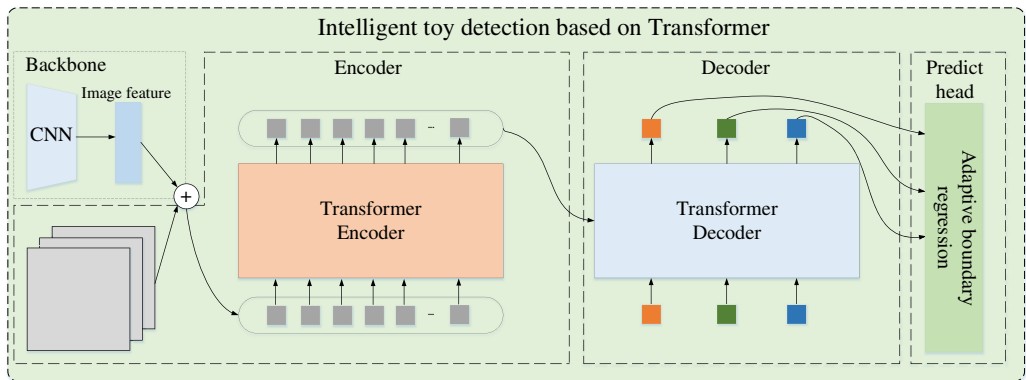

**Figure 1 Intelligent toy detection based on Transformer.**

more objects in a single computation due to its ability to handle large amounts of input data and parameters. This gives the Transformer a significant advantage in handling complex scenes, especially in applications that require detecting multiple objects and processing high-resolution images.

The input image generates a multi-dimensional original feature map through the convolutional neural network (CNN) backbone network and then converts the multi-dimensional original feature map into a one-dimensional feature map. Combined with the image position coding, the Transformer encoder outputs fixed-length vectors. The decoder is fed with the target query. The adaptive boundary regression model processes the decoder output to get the associated target region and the intelligent toy region.

The adaptive boundary representation of the intelligent toy positioning region is based on the prior shape of the toy, and the accurate toy positioning region is obtained by fitting the coordinates of its boundary points through the adaptive boundary regression model (ABRM). The adaptive boundary regression model is shown in Fig. 2. The text area boundary suggestion box is obtained through RPN and ROI. Then, the boundary suggestion box is optimized to obtain the accurate boundary box for the specific associated target and intelligent toy in the scene. The boundary points of the boundary suggestion box are expressed as $(x_1, y_1, x_2, y_2)\ldots\ldots(x_i, y_i, x_{i+1}, y_{i+1})$. In the inference stage, the network model will adaptively output the optimal number of boundary points according to the prior shape of the toy. By incorporating a refined region-adaptive boundary representation technique, the model can accurately capture the position and boundaries of toys in images, thereby achieving efficient localization of smart toys.

The loss of each suggestion box is defined as classifying loss, border regression loss and boundary point regression loss. The loss is as follows:

$$L = L_{Cls}(p, t) + t \sum_{i \in x_i, y_i, x_{i+1}, y_{i+1}} L_{reg}(v_i, v_i^*) + (t-1) \sum_{i \in x_i, y_i, x_{i+1}, y_{i+1}} L_{reg}(u_i, u_i^*) \qquad (1)$$

where the toy area loss $L_{cls}(p, t) = -\log p_t$. The $t$ represents the classification label. When $t = 1$, it is a toy area; when $t = 0$, it is not a toy area. The argument $p = (p_0, p_1)$ is the confidence of the toy zone and the non-toy zone calculated by softmax.

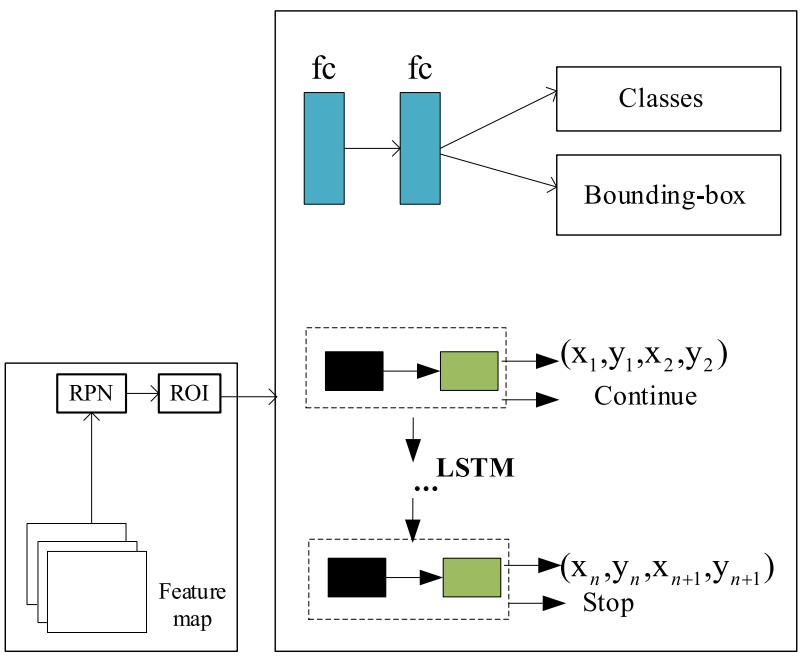

**Figure 2 Adaptive boundary regression model.**

The border regression loss and the boundary point regression loss can be expressed in a uniform form $(w, w^*)$, $L_{reg}(w, w^*)$ is presented as follows:

$$L_{reg}(w, w^*) = smooth_{L1}(w - w^*) \tag{2}$$

### Intelligent toy trajectory tracking based on depth-first search

After obtaining the toy location area of each frame image, we build the interrelation between frames. LSTM is used to determine the correlation of toy targets between different frames. LSTM mainly consists of a memory unit and three gating structures. Through the cooperative calculation of the three gating structures, the model can store and update the sequence information in the memory unit for a long time. A single neuron cell of LSTM is shown in Fig. 3. During the learning process, the weight that LSTM needs to update is memory cell ct. At time t, the input and output divisions of LSTM are defined as at and ht, then the input gate as i can be presented as follows:

$$i_t = \sigma(W_{ia}a_t + W_{ih}h_{t-1} + b) \tag{3}$$

where $W_{ia}$ and $W_{ih}$ are linear transformation matrices, the input gate controls the at and updates the memory cell $c_t$. The forgetting gate is the most characteristic structure of LSTM, which can control the data that memory cells need to forget to reduce the probability of being interfered with by unnecessary information in the following moments. The forgetting gate can also play a role in dimensionality reduction while discarding the

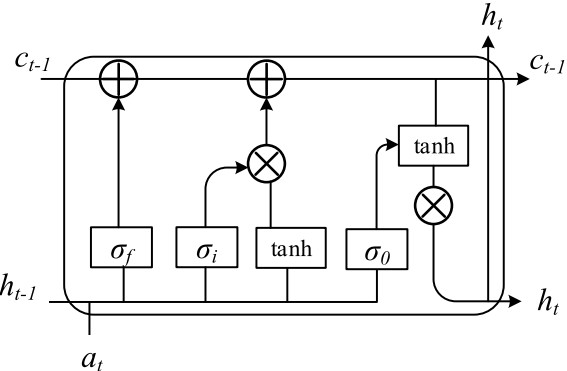

**Figure 3 The diagram of LSTM basic structure.**

memory information. Through the calculation of the forgetting gate, the memory cell $c_{t-1}$ of the previous moment will be updated to $c_t$, as shown in the formula:

$$f_t = \sigma\left(W_{fa}a_t + W_{fh}h_{t-1} + b\right) \tag{4}$$

$$c_t = f_t c_{t-1} + i_t \tanh(W_{ca}a_t + W_{ch}h_{t-1} + b) \tag{5}$$

Through the updated memory cell $c_t$, the output gate of the current moment $t$ is constructed. The output gate mainly calculates the influence $c_t$ on the output value $h_t$, as shown in the formula:

$$o_t = \sigma(W_{oa}a_t + W_{oh}h_{t-1} + b) \tag{6}$$

$$h_t = o_t \tanh(c_t) \tag{7}$$

where $W_{oa}$ and $W_{oh}$ are linear mapping functions.

However, the association between toys constructed by LSTM is limited to frame to frame, and it isn't easy to achieve multi-frame multi-toy target tracking. To solve the above problems, we embed a depth search algorithm in the loop process of LSTM to ensure the uniqueness of multiple targets in different frames.

First, we represent the toy targets across frames as a deep search graph (DS-Graph). The DS-Graph is divided into two layers: global and local relational graphs $G_k$. Subsequently, a deep search tree is constructed to facilitate cross-frame target search. The core idea is to establish an overall relationship between the subgraphs within each site based on the global S-Graph Σ. For each distributed site, we utilize the fundamental concept of recursive calls to preprocess the data subgraph $G_k(1 \leq k \leq p, k \neq i)$ within that site. This preprocessing involves computing the region's local S* -Graph and determining its various $T_i$ components. We then apply a merge algorithm to obtain the DFS tree of $G_k$, identifying leaf nodes. Using these leaf nodes, we search for the next $G_k$, repeating the process with unsearched leaf nodes as roots until no more leaf nodes can serve as roots, as illustrated in Fig. 4. Finally, we establish target chains using the relationships between the completed search leaf nodes and root nodes, enabling the tracking of multiple targets across multiple frames.

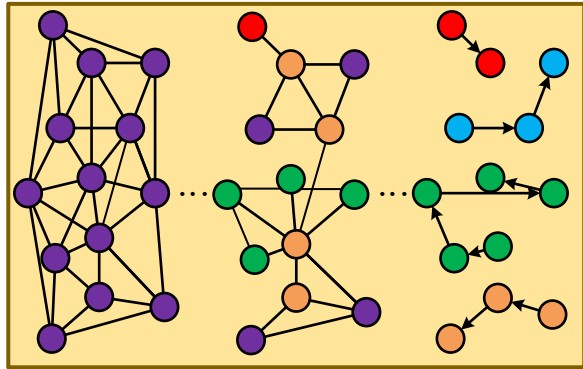

**Figure 4  The process of the deep search graph.**     

In toy tracking, LSTM can be used to process time series data related to toy motion. For example, by analyzing the historical position data of the toy, LSTM can learn the motion pattern of the toy and predict its future position. This predictive ability can be used to improve the accuracy and efficiency of toy tracking. At the same time, the depth-first search algorithm can be used to optimize graph search or path planning problems related to toy tracking. For example, when constructing a path map of possible movements for the toy, DFS can be used to search for all possible paths. By integrating the DFS strategy with the outputs of deep learning networks, we can significantly enhance the performance of target tracking. The deep learning network not only provides precise feature representations and initial position predictions of the target but also, combined with the DFS strategy, ensures that all potential target locations or states are thoroughly explored, leading to the most accurate tracking results. During the tracking process, we start from the initial frame of the video and utilize the target feature representations extracted by the deep learning network to search for the most similar location or state of the target in each subsequent frame. If a satisfactory result is not found in a particular frame, we leverage the backtracking capability of DFS to return to previous frames and explore other possible paths. Through this continuous process of searching and backtracking, we can gradually narrow down the search range and precisely locate the accurate position or state of the target across different frames. This approach improves the accuracy of target tracking and enhances its robustness in complex scenarios.

## Model lightweight deployment based on mobile cloud terminal

To realize the transplantation of the proposed method in the intelligent toy terminal, we must realize its realtime and portability. We propose a lightweight model deployment method based on mobile cloud terminals. Mobile cloud terminals provide great convenience for information resource access and collaborative work. Still, their computing performance and power supply can not meet the needs of many screen rendering and complex computing tasks for smart toys. Mobile terminals formulate computing migration strategies according to network delay, bandwidth and energy consumption, and local process tasks with high latency requirements to improve user experience. Migrate tasks that consume too much computing resources to cloud platform services.

When the mobile cloud terminal assigns the task to the cloud for processing, the model is not lightweight. When the mobile cloud terminal assigns tasks to the cloud for local processing, we will lighten the proposed model. The Transformer-based smart toy detection model is computationally intensive in the entire tracking approach. Therefore, we reduce and optimize its structure. For individual attention modules, we use Transformer architecture similar to Segformer (*Xie, Wang & Yu, 2021*). Reducing the key-value sequence on a single scale is not conducive to segmenting small objects. Therefore, we use shunt self-attention (SSA) to improve the segmentation. SSA obtains the context interrelationships of multiple scales of the target by reducing the key-value sequence at multiple scales. SSA is grouped by header, and different header key-value sequences are scaled differently. Assuming i is the index of the header, then SSA can be represented as:

$$Q_i = X_k W_i^Q \tag{8}$$

$$K_i = DS(X_k, r_i) W_i^K \tag{9}$$

$$V_i = DS(X_k, r_i) W_i^V \tag{10}$$

$$V_i = V_i + LE(V_i) \tag{11}$$

where $X_k$ represents the input sequence, and $Q_i$, $K_i$, and $V_i$ refer to the query, key and value features mapped from the $i$-th header, respectively. All $W$ are trainable linear transformation matrices. DS($\cdot$) denotes the downsampling of the $i$-th head, and $LE(\cdot)$ refers to the enhancement operation on $V_i$. Self-attention is lightened by the above method, leaving the rest of the calculations unchanged.

The lightweight deployment of the smart toy tracking model enables it to run more efficiently on mobile devices, reducing delays and stuttering and providing users with a smoother and more natural interactive experience. The model undergoes lightweight processing to reduce data volume and computational complexity, lowering the hardware requirements for mobile devices. The smart toy tracking model requires realtime tracking of the toy's position and status to allow users to control and interact accurately. Lightweight deployment reduces the time cost of model operation and improves realtime performance while maintaining high tracking accuracy, ensuring that users receive timely and accurate information feedback.

In summary, the scalability issue has been effectively addressed through comprehensive optimization measures such as algorithm optimization, model lightweighting, data processing, system architecture improvements, and scalability testing. These measures enable the system to maintain efficient and stable operation in large-scale data and complex environments.

## EXPERIMENT AND ANALYSIS

### Dataset and implement details

We use the video from the Trajectory Robot Dataset (https://www.zenodo.org/record/6337847, 10.5281/zenodo.6337847) to test an intelligent toy trajectory tracking algorithm based on mobile cloud deployment and depth-first search. The dataset comprises color and depth videos capturing the movements of the Panda robot, along with their respective joint

**Table 1 Implementation details.**

| Parameters | Value |
|---|---|
| Initial learning rate | $4 \times 10^{-4}$ |
| Epoch | 40 |
| Batch-size | 10 |
| Decay | 0.9 |
| Gradient descent mode | Adam |
| Image input size | $420 \times 420$ |
| Image feature dimensions | 512 |

and Cartesian trajectories. Additionally, it encompasses the trajectories of a receiver robot involved in object handover. Each motion instance encompasses six files: RGB video, depth video, and four trajectories (pertaining to the giver and receiver) presented in a time series format. The dataset encompasses a total of 38,393 motion samples. This dataset comprises video data captured in various real-world scenarios, encompassing diverse lighting conditions, weather situations, and object movement patterns. It offers training data for multi-object tracking, assisting robots in learning how to manage tracking tasks for multiple targets efficiently. By training on this diverse dataset, video trajectory robot models can better adapt to various real-world scenarios, enhancing their generalization capabilities.

Since the Transformer model requires extensive computational resources for training on large datasets, this chapter leverages the robust computing capabilities of the calibration supercomputer center, equipped with CPUs (Xeon(R) E5-2640 v4) and GPUs ($4 \times$ Nvidia Tesla V100), to facilitate the setup of the environment and subsequent model training. The chosen deep learning framework remains Pytorch. To adapt to the cross-modal training task of the Transformer, the experiment pre-trained Faster R-CNN. Unlike image grid features, the method in this chapter needs to input accurate target category and location information. Therefore, datasets Objects365, MSCOCO, OpenImages, and Visual Genome were employed to complete the pre-training of the object detection model. Experimental parameters are presented in Table 1.

Considering that intelligent toy track tracking is a task to judge successive frames, we employ mean square error (mAP) and F-value as the evaluation criteria of the method, which can be as follows:

$$Precision = \frac{V(gt \bigcap pr)}{V(pr)} \tag{12}$$

$$Recall = \frac{V(gt \bigcap pr)}{V(gt)} \tag{13}$$

$$F = \frac{2 \times Precision \times Recall}{Precision + Recall} \tag{14}$$

$$mAP = \frac{1}{N} \times \sum Precision_n \times Recall_n \tag{15}$$

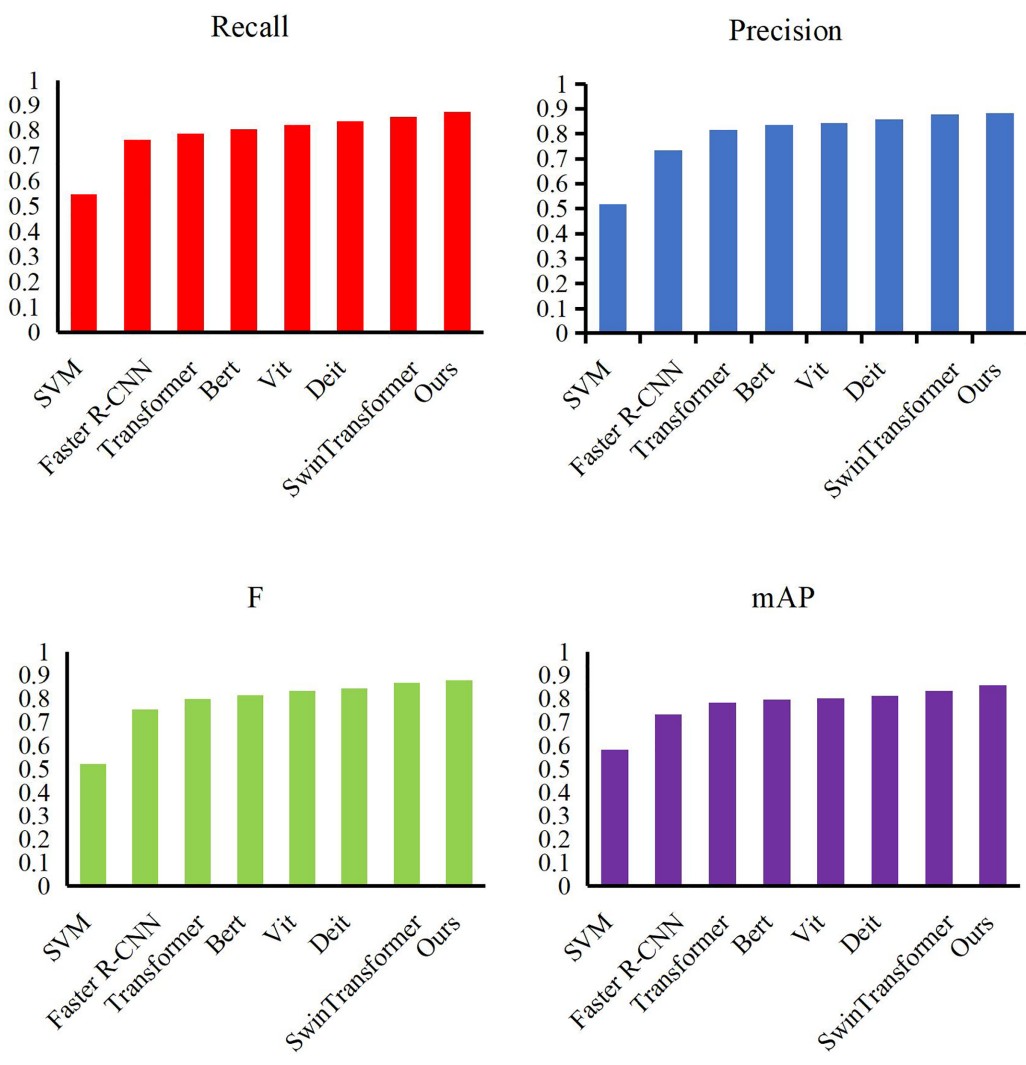

**Figure 5  Compare our detection method with other methods.**

where pr denotes the result from the model, and gt refers to the ground truth. In the context of smart toy tracking, the rationale for selecting mAP and F-value as evaluation metrics lies in their ability to comprehensively and accurately reflect the precision and recall of the model in detecting multiple target toys, thereby effectively assessing the model's performance.

## Compare our detection method with other methods

We conducted an experiment using our method on the Video-the Trajectory Robot Dataset dataset. We selected some excellent feature models, such as SVM (*Chauhan, Dahiya & Sharma, 2019*), Faster R-CNN, Transformer (*Vaswani, Shazeer & Parmar, 2017*), Bert (*Deepa, 2021*), Vit (*Khan, Naseer & Hayat, 2022*), Deit (*Touvron, Cord & Douze, 2021*) and SwinTransformer (*Liu, Lin & Cao, 2021*), and compared their performance. We present the results in Fig. 5 and Table 2. While comparing with other

**Table 2 Compare the proposed method with others.**

| Methods | Recall | Precision | F | mAP |
|---|---|---|---|---|
| SVM | 0.548 | 0.517 | 0.523 | 0.584 |
| Faster R-CNN | 0.763 | 0.734 | 0.753 | 0.732 |
| Transformer | 0.786 | 0.814 | 0.799 | 0.783 |
| Bert | 0.804 | 0.834 | 0.816 | 0.795 |
| Vit | 0.821 | 0.843 | 0.832 | 0.801 |
| Deit | 0.837 | 0.856 | 0.845 | 0.812 |
| SwinTransformer | 0.854 | 0.876 | 0.867 | 0.834 |
| Ours | 0.873 | 0.883 | 0.879 | 0.858 |

target location methods, our method achieves the highest values among all evaluation indexes, namely 0.854 recall, 0.876 precision, 0.867 F-value and 0.834 mAP. Compared with SVM, our method increases the mAP value by more than 27%, mainly because SVM is difficult to achieve the fitting of large-scale data sets. Compared to Faster R-CNN, our method has more than 12% lead. Unlike SVM, Faster R-CNN can achieve rapid convergence but cannot completely avoid the interference caused by complex backgrounds. Self-attention-based Transformer and Bert can handle detection tasks in complex environments, and our approach still improves mAP scores by more than 5.5%. Vit and Deit are the most advanced detection methods, and they can achieve more than 80% mAP value thanks to excellent model performance. Our method achieves a lead of more than 4% compared to this. Finally, our approach is also completely ahead of the curve compared to Swin Transformer. Our approach uses an adaptive candidate box mechanism to help the model overcome the error detection caused by complex scenes to obtain more efficient detection results.

In addition, according to the method's structure, our training is compared with other models. The loss curve of the model is in Fig. 6. It can be seen that ours reaches the fit at epoch 35. On the other hand, the model usually fits at epoch 40. In addition, compared with Bert, Vit and Swin Transformer, we can conclude that our model can converge earlier and the training process is smoother.

## Compare our tracking method with other methods

At the end of the validation of our object detection method, we evaluate the performance of our trajectory tracking model with Expect Average Overlaprate (EAO) and Multiple Object Tracking Precision (MOTP). We compare our method with some excellent models, such as DeepSORT (*Veeramani, Raymond & Chanda, 2018*), FlowTrack (*Zhu, Wu & Zou, 2018*), CenterTrack (*Zhou, Koltun & Krahenbuhl, 2020*) and Spatio-temporal Transformer (*Yan, Peng & Fu, 2021*). In Table 3, Figs. 7 and 8, our method has demonstrated outstanding performance in comparison with other tracking algorithms, achieving the highest scores across all evaluation metrics, specifically an EAO score of 0.351, a MOTP score of 0.885, and a MOTA score of 0.916. Compared to DeepSORT, our method significantly improves the EAO score by 3.9% and the MOTP score by 6.2%. Compared to

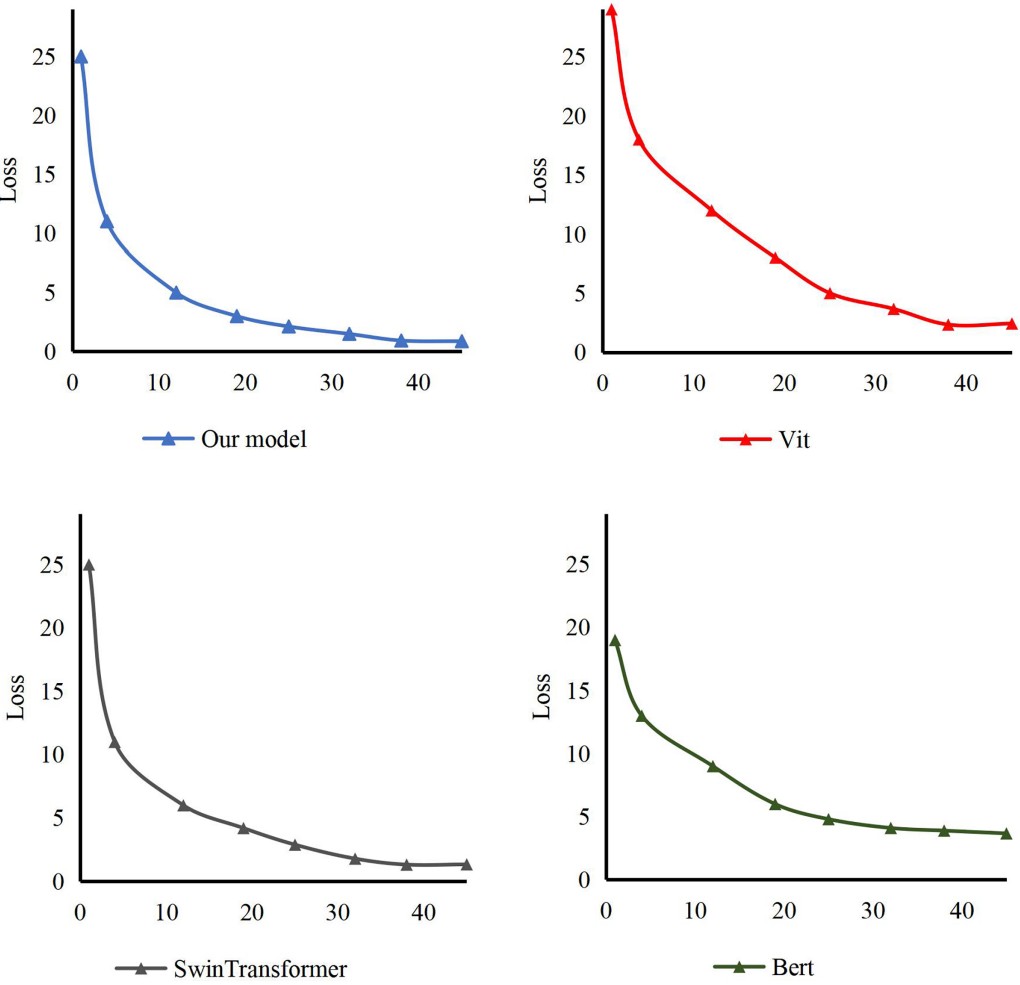

**Figure 6** **The loss of our model comparing with others.**

**Table 3 Compare our tracking method with other methods.**

| Methods | EAO | MOTP | MOTA |
|---|---|---|---|
| DeepSORT | 0.312 | 0.823 | 0.867 |
| FlowTrack | 0.345 | 0.831 | 0.879 |
| CenterTrack | 0.324 | 0.856 | 0.896 |
| Spatio-temporal transformer | 0.338 | 0.867 | 0.902 |
| Ours | 0.351 | 0.885 | 0.916 |

FlowTrack, we achieve a lead of over 5% in the MOTP score, albeit with a slightly higher inference time and model parameters. However, this minor cost results in significant performance gains. Compared to CenterTrack, our method takes the lead across all metrics, outperforming it in every aspect, requiring fewer model parameters and a shorter inference time. This comprehensive advantage is primarily attributed to integrating the deep search mechanism, which significantly enhances the performance of our method.

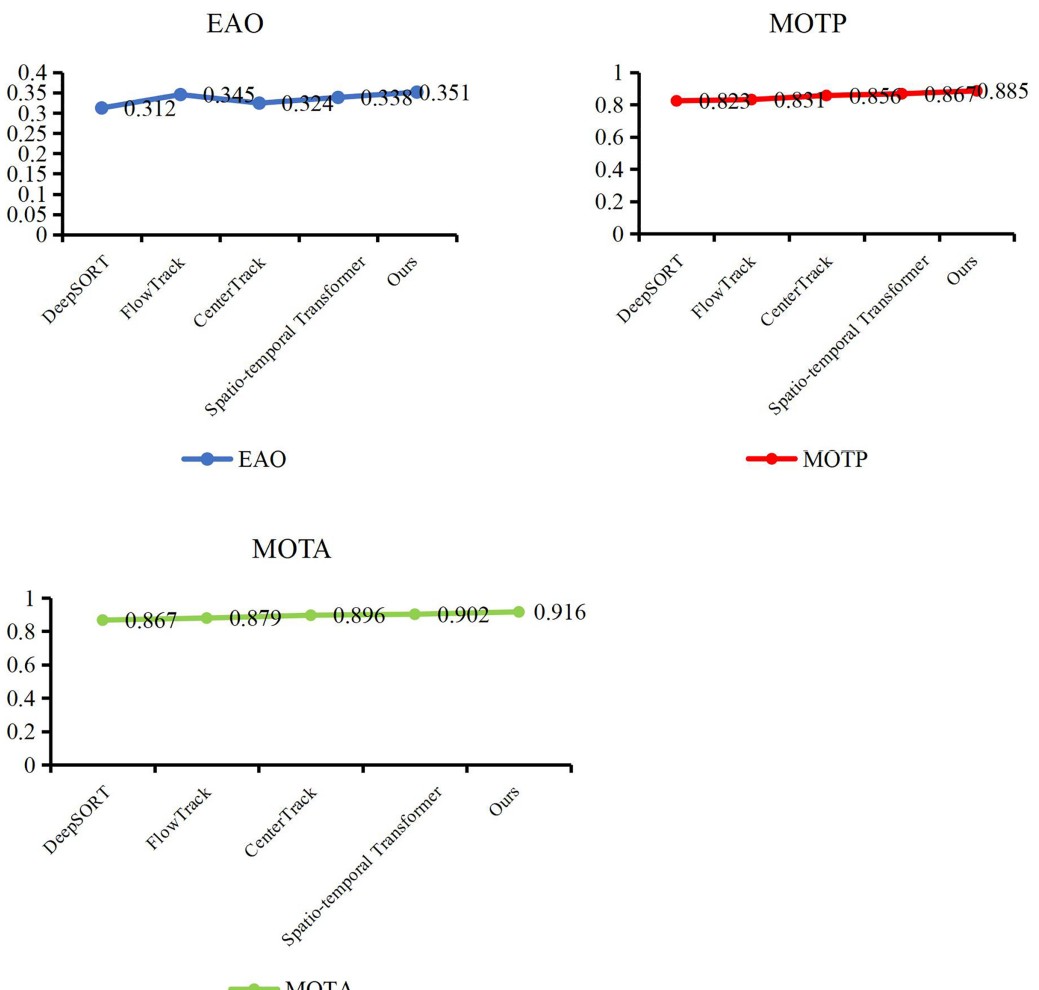

**Figure 7 Compare our tracking method with other methods.**

Finally, compared with the spatiotemporal transformer, our method achieves improvements of 1.3% in EAO score, 1.8% in the MOTP score, and 1.4% in the MOTA score. This is due to our utilization of a structurally simple LSTM in constructing the tracking method, coupled with the further enhancement of LSTM performance through a deep search algorithm, which enables us to maintain performance advantages while reducing the model's data size. By leveraging deep learning networks and incorporating the concept of depth-first search, a model can explore potential locations or states of a target across different video frames and then combine this information with the outputs of a neural network. Experiments have shown that through continuous searching and backtracking, we can gradually narrow down the search space and ultimately identify the accurate position or state of the target in different frames.

## Ablation experiments

We conduct ablation experiments on a dataset focusing on three sub-modules: Transformer, LSTM, and DFS, to evaluate their impacts on model performance. Initially,

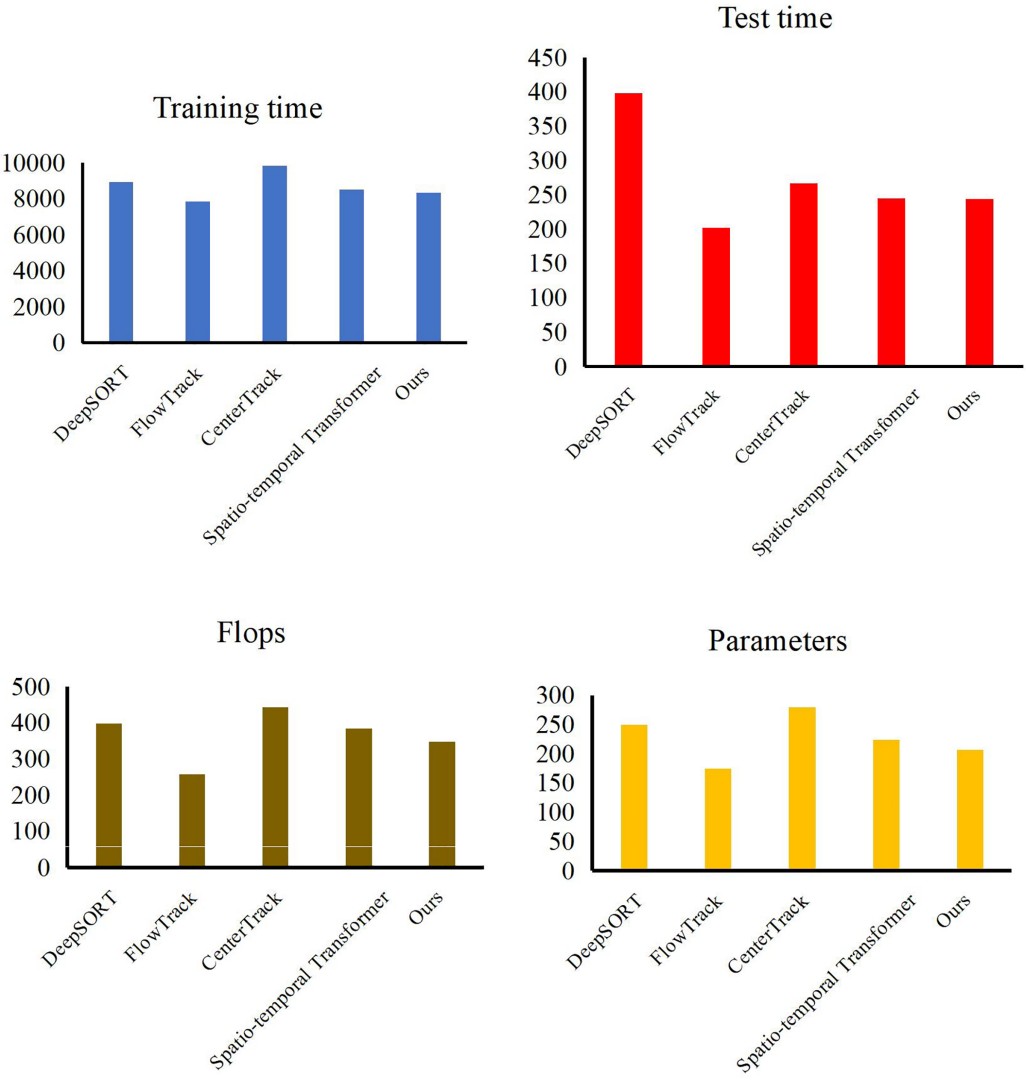

**Figure 8 Model efficiency comparison with other methods.**

we observe the performance of the baseline model, which achieves an EAO (Expected Average Overlap) score of 0.256 and a MOTP (Multiple Object Tracking Precision) score of 0.792, as seen in Table 4. Subsequently, we embed the Transformer, LSTM, and DFS modules into the baseline model separately to explore their contributions to improving the model's performance. The experimental results show that embedding the Transformer increases the EAO score to 0.298, embedding the LSTM raises it to 0.304, and embedding the DFS further enhances it to 0.319. These results strongly demonstrate the effectiveness of these three modules in enhancing model performance. To further investigate the combined effects of these modules, we conduct ablation experiments with pairwise combinations. The experimental results reveal that when combining Transformer and LSTM, the model achieves a MOTP score of 0.859; when combining Transformer and DFS, the MOTP score reaches 0.865; and when combining LSTM and DFS, the MOTP

**Table 4  Ablation experiments.**

| Transformer | LSTM | DFS | EAO | MOTP |
|---|---|---|---|---|
| | | | 0.256 | 0.792 |
| O | | | 0.298 | 0.831 |
| | O | | 0.304 | 0.829 |
| | | O | 0.319 | 0.811 |
| O | O | | 0.338 | 0.859 |
| O | | O | 0.328 | 0.865 |
| | O | O | 0.341 | 0.868 |
| O | O | O | 0.351 | 0.885 |

score attains 0.868. These findings further illustrate the complementarity between different modules and their synergistic effects on different tasks. Finally, we simultaneously embed all three modules—Transformer, LSTM, and DFS—into the baseline model to explore their optimal performance when combined. The experimental results indicate that the combination of these three modules achieves an outstanding MOTP score of 0.885. This result validates the respective advantages of these three modules and showcases the powerful synergistic effect they can produce when combined.

## CONCLUSION

To improve the playability and portability of smart toys, we propose an intelligent toy tracking model by the mobile cloud terminal deployment and depth-first search algorithm. An intelligent toy detection model *via* a Transformer is proposed to improve the toy detection effect of a single frame. LSTM and embedded depth-first search mechanisms are used to realize the trajectory tracking of intelligent toys in successive frames. In addition, to boost the portability of our method, a lightweight model deployment method based on a mobile cloud terminal is proposed to realize the multi-functional application of smart toys. Experiments can demonstrate that our proposed toy positioning method can obtain a mAP value of 0.858 and achieve accurate detection within a single frame. Our continuous frame toy tracking method can obtain the MOTA value of 0.916, which realizes the real-time tracking function of intelligent toys.

### Funding
The authors received no funding for this work.

### Competing Interests
Both authors are employed by Beijing AIQI Technology Co., LTD, The authors declare that they have no competing interests.

### Author Contributions
- Yang Zhang conceived and designed the experiments, analyzed the data, performed the computation work, prepared figures and/or tables, and approved the final draft.

- Hu Zhang performed the experiments, analyzed the data, authored or reviewed drafts of the article, and approved the final draft.

## Data Availability

The code is available in the Supplemental File.

The dataset is available at Zenodo: Mavsar, M. (2022). Video-Trajectory Robot Dataset [Data set]. Zenodo. https://doi.org/10.5281/zenodo.6337847.

## Supplemental Information

Supplemental information for this article can be found online at http://dx.doi.org/10.7717/peerj-cs.2187#supplemental-information.

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
