# Peer review of "Intelligent toy tracking trajectory design based on mobile cloud terminal deployment and depth-first search algorithm"

_PeerJ Computer Science, doi:10.7717/peerj-cs.2187_

## Round 0.1 · original submission · Major Revisions

The reviewers have now commented on the technical quality of the paper and you will see that there are many concerns for improving your article, they have been outlined below. I do agree with them and therefore, endorse their comments. so please carefully revise the paper in light of the reviewers' comments and the comments from my side below.

You need to resubmit a detailed response along with an updated manuscript responding to the point-by-point responses to the reviewers and Academic Editor.

AE Comments:

Provide more context about the significance and current state of intelligent toys in the market. Why is the proposed tracking method necessary?

Clarify what makes the proposed method novel compared to existing solutions. Highlight the specific problems it addresses in a separate paragraph or section.

Explain the key terms and acronyms (e.g., Transformer, LSTM, mAP, MOTA) for readers who may not be familiar with them

Explain the concept of “refined region adaptive boundary representation” in more detail. How does it improve toy detection?

Summarize the key contributions of the proposed method and its potential impact on the field of intelligent toys

Reviewer 1 ·

Basic reporting

Please find below my comments on your manuscript. These are intended to enhance the clarity, accuracy, and academic rigor of your work.

Abstract, Sentence: "Firstly, we construct a toy detection model via Transformer which realizes the positioning of toys in the image through the refined region adaptive boundary representation." Specify the novelty of your Transformer model compared to existing models for toy detection.

The metrics used to evaluate model performance (e.g., mAP, F-value) are appropriate, but the paper should include more detailed definitions and justification for their selection in the context of intelligent toy tracking.

The choice of the Video the Trajectory Robot Dataset is sound, yet further elaboration on how this dataset specifically addresses real-world scenarios and challenges would be valuable.

Section 3.2 would benefit from additional technical details regarding the integration of the depth-first search algorithm with LSTM, particularly how this improves tracking accuracy and efficiency.

A more comprehensive description of the experimental setup, including hardware specifications, training parameters, and environmental conditions, would enhance reproducibility.

The graphs presented, such as Figures 7 and 8, should include more detailed captions and interpretations, explaining the significance of the observed trends and anomalies.

Section 4.3, Sentence: "Compared to DeepSORT, our method improved EAO score by 3.9% and MOTP score by 6.2%." Suggest including a table summarizing these comparative performance metrics for easier reference.

Experimental design

Ensure that all formulas and equations are clearly defined and explained, particularly those that are central to the proposed method. Including a step-by-step derivation can enhance understanding.

Validity of the findings

Valid.

Additional comments

The clarity of the graphs can be improved by ensuring all axes are labeled correctly, units are provided where necessary, and legends are included for better interpretability.

·

Basic reporting

see below

Experimental design

see below

Validity of the findings

see below

Additional comments

I have reviewed your manuscript titled "Intelligent Toy Tracking Trajectory Design Based on Mobile Cloud Terminal Deployment and Depth-First Search Algorithm." The following comments are necessary to enhance the impact of your work:

The integration of Transformer, LSTM, and Depth-First Search is promising. However, a clearer delineation of each component's role and its synergistic effects would be beneficial.

Provide a more rigorous justification for the choice of depth-first search over other search algorithms, supported by comparative performance data.

The construction of the DS-Graph and its role in multi-frame tracking could be elucidated further, particularly in terms of computational complexity and scalability.

While comparing with methods such as DeepSORT and CenterTrack, a detailed analysis of the scenarios where your method outperforms these techniques would provide deeper insights.

Conducting ablation studies to isolate the impact of each component (e.g., SSA, LSTM, depth-first search) on overall performance would add robustness to your findings.

Figure 4, Caption: "The process of the deep search graph." Enhance the caption to explain the steps depicted in the graph and their significance in the context of your study.

Discuss the practical implications of deploying your model in real-world settings, addressing potential challenges and how your approach mitigates them.

Scalability and Adaptability: Address the scalability of your method to different types of intelligent toys and varying environmental conditions, highlighting any necessary adaptations.

---

## Round 0.2 · accepted · Accept

Thanks you for revision in light of the comments of reviewers and mine. Based on their feedback I'm pleased to inform you that your manuscript is in good condition to accept.
Good luck

Reviewer 1 ·

Basic reporting

The authors have revised the paper according to suggestions.

Experimental design

The authors have revised a comprehensive description of the experimental setup, including hardware specifications, training parameters, and environmental conditions, which would enhance reproducibility.

Validity of the findings

Valid

Additional comments

Please read carefully to improve minor grammatical mistakes.

·

Basic reporting

Accepted

Experimental design

Accepted

Validity of the findings

Accepted

Additional comments

Accepted